# Carryover Contamination-Controlled Amplicon Sequencing Workflow for Accurate Qualitative and Quantitative Detection of Pathogens: a Case Study on SARS-CoV-2

Lifen Gao,[a] Lun Li,[a] Bin Fang,[b] Zhiwei Fang,[a] Yanghai Xiang,[c] Min Zhang,[c] Junfei Zhou,[a] Huiyin Song,[a] Lihong Chen,[a] Tiantian Li,[a] Huafeng Xiao,[a] Renjing Wan,[a] Yongzhong Jiang,[b] Hai Peng[a,d]

[a]Institute for Systems Biology, Jianghan University, Wuhan, Hubei, People's Republic of China
[b]Hubei Provincial Centers for Disease Control and Prevention, Wuhan, Hubei, People's Republic of China
[c]Yueyang Central Hospital, Yueyang, Hunan, People's Republic of China
[d]Mingliao Biotechnology Co., Ltd., Wuhan, Hubei, People's Republic of China

Lifen Gao, Lun Li, and Bin Fang contributed equally to the manuscript. Author order was determined on the basis of their contributions.

**ABSTRACT** Carryover contamination during amplicon sequencing workflow (AMP-Seq) put the accuracy of the high-throughput detection for pathogens at risk. The purpose of this study is to develop a carryover contaminations-controlled AMP-Seq (ccAMP-Seq) workflow to enable accurate qualitative and quantitative detection for pathogens. By using the AMP-Seq workflow to detect SARS-CoV-2, Aerosols, reagents and pipettes were identified as potential sources of contaminations and ccAMP-Seq was then developed. ccAMP-Seq used filter tips and physically isolation of experimental steps to avoid cross contamination, synthetic DNA spike-ins to compete with contaminations and quantify SARS-CoV-2, dUTP/uracil DNA glycosylase system to digest the carryover contaminations, and a new data analysis procedure to remove the sequencing reads from contaminations. Compared to AMP-Seq, the contamination level of ccAMP-Seq was at least 22-folds lower and the detection limit was also about an order of magnitude lower—as low as one copy/reaction. By testing the dilution series of SARS-CoV-2 nucleic acid standard, ccAMP-Seq showed 100% sensitivity and specificity. The high sensitivity of ccAMP-Seq was further confirmed by the detection of SARS-CoV-2 from 62 clinical samples. The consistency between qPCR and ccAMP-Seq was 100% for all the 53 qPCR-positive clinical samples. Seven qPCR-negative clinical samples were found to be positive by ccAMP-Seq, which was confirmed by extra qPCR tests on subsequent samples from the same patients. This study presents a carryover contamination-controlled, accurate qualitative and quantitative amplicon sequencing workflow that addresses the critical problem of pathogen detection for infectious diseases.

**IMPORTANCE** Accuracy, a key indicator of pathogen detection technology, is compromised by carryover contamination in the amplicon sequencing workflow. Taking the detection of SARS-CoV-2 as case, this study presents a new carryover contamination-controlled amplicon sequencing workflow. The new workflow significantly reduces the degree of contamination in the workflow, thereby significantly improving the accuracy and sensitivity of the SARS-CoV-2 detection and empowering the ability of quantitative detection. More importantly, the use of the new workflow is simple and economical. Therefore, the results of this study can be easily applied to other microorganism, which has great significance for improving the detection level of microorganism.

**KEYWORDS** carryover contamination, amplicon sequencing, synthetic DNA spike-ins, dUTP/UDG, SARS-CoV-2

Address correspondence to Hai Peng, penghai138@163.com, or Yongzhong Jiang, hbcdcxd@163.com.

The authors declare no conflict of interest.

Amplicon sequencing offered a highly simple and high-throughput method for the detection of infectious disease pathogens and has been widely used to detect pathogens for research and clinical diagnostics (1–8). The common approaches to generate amplicon libraries suitable for next-generation sequencing includes the first step of PCR amplification of the target gene, followed by the second step of adding sample barcodes and sequencing adapters to the PCR products by PCR or ligation reaction. Typically, hundreds or thousands of gene sequences are amplified per reaction via multiplexed PCR (9, 10). This makes it sensitive to the detection of pathogens but also highly susceptible to contaminations by the amplicons carried over from the prior steps, thereby causing false-positive and false-negative errors. False-positive tests cause harm to the individuals, and false-negative tests might result in inadequate precautions and increased disease transmission. Carryover contamination has been reported in PCR-based nucleic acid diagnostic techniques (11–15). Different sets of primer combinations have been used to monitor carryover contamination during amplicon sequencing (15); however, this strategy is expensive as it requires synthesis of multiple sets of multiplex primers and cannot remove contamination. The dUTP/thermolabile uracil DNA glycosylase (UDG) system has been used to eliminate contaminations during RT-LAMP and qRT-PCR workflow by supplementing PCR mixtures with dUTP and UDG (14, 16). The carryover amplions were cleaved by UDG enzyme before PCRs at the dUTP nucleotides. However, comprehensive studies on the generation and prevention/elimination of carryover contamination from amplicon sequencing system are lacking.

In the present study, we developed a conventional amplicon sequencing (AMP-Seq) workflow for the detection of 164 multiple dispersed nucleotides polymorphism (MNP) markers (17) in the SARS-CoV-2 genomes, which can be used to determine contamination occurring during the amplicon sequencing workflow. We comprehensively evaluated contamination at steps of library construction and strategies to prevent/eliminate carryover contamination. We subsequently established a carryover contamination-controlled amplicon sequencing workflow ccAMP-Seq and associated data analysis pipeline. Using gradient dilutions of SARS-CoV-2 nucleic acid standards and 62 clinical samples, the superiority of ccAMP-Seq workflow over AMP-Seq workflow for SARS-CoV-2 detection on accuracy and quantification were demonstrated.

## RESULTS

**The overview of works for developing contamination-controlled amplicon sequencing workflow ccAMP-Seq.** First, we developed an amplicon sequencing workflow (AMP-Seq) for SARS-CoV-2 detection using a two-step PCR strategy for library construction (Fig. 1A). In the use of AMP-Seq for SARS-CoV-2 detection, we found that an increasing number of SARS-CoV-2 reads were detected in the nontarget control samples (NTCs) consisting of nuclease-free and sterile water (NFS water). Then we analyzed the possible sources of carryover contamination, including aerosols, reagents, and equipment for PCR during the workflow. Subsequently, we developed ccAMP, including the use of filter tips, the addition of synthetic DNA spike-ins and dUTP/UDG system to prevent/eliminate contaminations during library construction and the establishment of a new data analysis procedure to remove the sequencing reads from contaminations (Fig. 1B).

**The possible sources of carryover contamination in AMP-Seq workflow.** To evaluate the possible aerosols contamination in AMP-Seq workflow, we tested the NFS water that had been placed in the PCR preparation and analysis rooms with lid open for 1 day and 1 week, and outdoor away from the laboratory with lid open for 1 day, using AMP-Seq. At the week for sample preparation, routine molecular testing other than SARS-CoV-2 was conducted in these rooms as usual. Each condition performed one test and all tests were performed in laboratories with physical separation for experimental steps (Samples 1 to 5 in Supplemental file 1: Table S2). The target value (T value) for each sample, which is the ratio of the number of reads mapped to SARS-CoV-2 MNP loci to the number of total qualifying reads of the sample, was used to measure the contamination levels in the samples. The T values of the samples placed

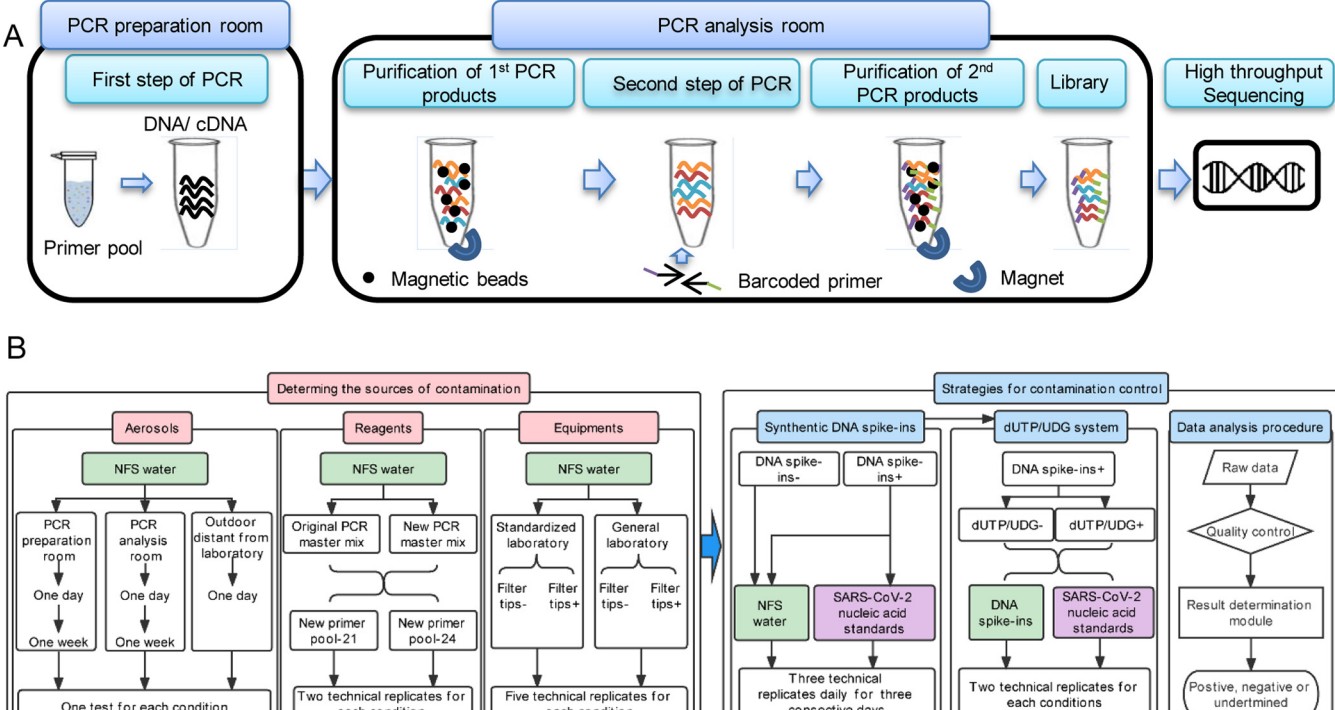

**FIG 1** The overview of works for developing contamination-controlled workflow ccAMP-Seq. (A) The established amplicon sequencing workflow AMP-Seq used a two-step PCR strategy for library construction. (B) The procedure for developing ccAMP-Seq involved the analysis of possible sources of contamination during AMP-Seq and the development and evaluation of strategies to prevent/eliminate contamination. NFS water: nontarget control; Standardized and general laboratories: laboratories those were physically isolated and nonphysically isolated from each step of library construction.

in the PCR preparation and analysis room for 1 day (0.36% and 0.32%, respectively) were higher than the samples placed for 1 week (0.21% and 0.19%, respectively), suggesting that aerosol contamination was present in all these rooms and dissipated over time. However, T value (0.31%) of the NFS water sample placed outdoors far from the laboratory was similar to those in the PCR preparation and analysis rooms, indicating possible contamination of the reagents or equipment used for library construction (Fig. 2A). We then resynthesized the primer pools containing 21 and 24 of the 164 pairs to test newly purchased NFS water samples using newly purchased and original PCR master mix reagents (Samples 6 to 13 in Supplemental file 1: Table S2). Each condition tested two technical replicates. Samples tested using the new PCR master mix had lower T values (0.01% on mean) than those tested with the original mix (9.18% on mean), confirming contamination of reagents. The large fluctuations of T values between replicates (17.99% versus 3.22% and 12.81% versus 2.72%) tested by both the two new primer pools using original PCR master mix indicated the randomness of contamination (Fig. 2B).

We next tested NFS water in laboratories that were physically isolated and nonphysically isolated from each step of library construction (standardized and general laboratories, respectively). Pipettes were also used with and without filter tips (Samples 14 to 33 in Supplemental file 1: Table S2). Each condition tested five technical replicates. The mean T value of samples tested using filter tips in standardized laboratories (0.43%) was significantly lower than that of samples tested not using filter tips (1.12%) and samples tested using and not using filter tips in general laboratories (0.97% and 1.28%, respectively) (Wilcoxon rank-sum test, $P < 0.05$; Fig. 2C), indicating contamination of pipettes used for library construction.

**Effective strategies for carryover contamination control. (i) Synthetic DNA spike-ins.** Filter tips and standardized laboratories can be used to physically isolate contamination. Once contamination enters the PCR, additional strategies are required. In this study, we designed and synthesized 17 SARS-CoV-2 MNP loci-derived fragments

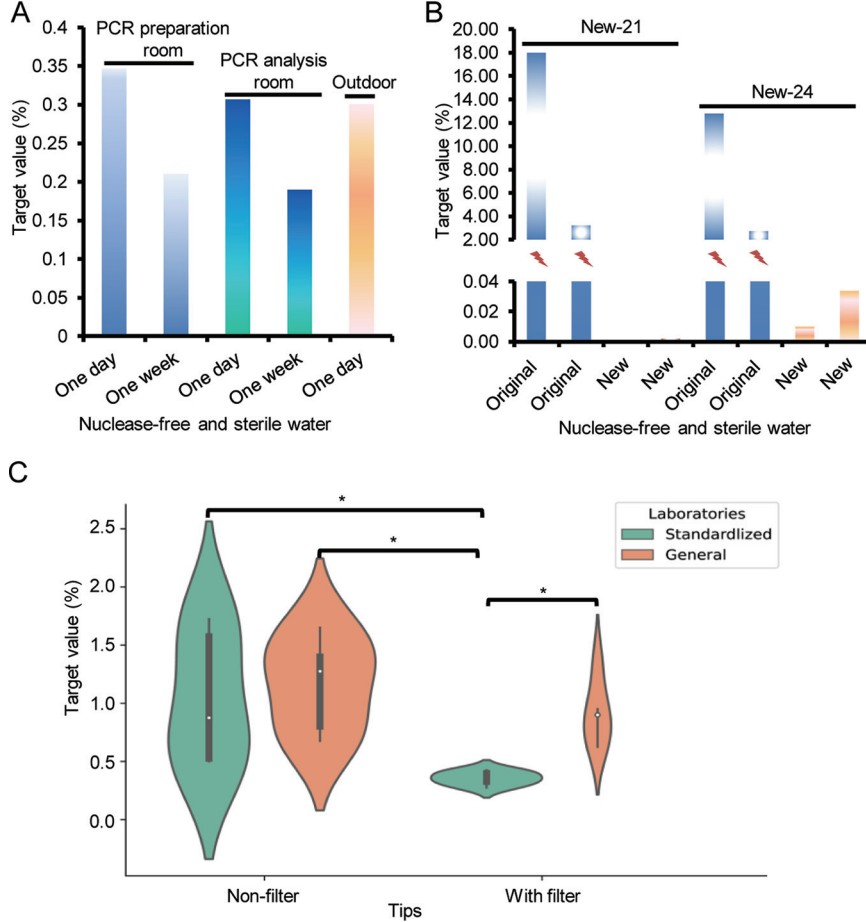

**FIG 2** The possible sources of contaminants. (A) Aerosol contamination was evaluated by analyzing Nuclease-free and sterile water (NFS water) placed in the PCR preparation and analysis room for 1 day and 1 week with the lid open, and in the outdoor 1 day with the lid open. (B) Contamination in the PCR components, including the reagents and primers, was evaluated by analyzing NFS water using newly synthesized primer pools and newly purchased master mix. (C) Contamination in the equipment for PCR such as pipettes was evaluated by analyzing NFS water using tips with and without filter in standardized and general laboratories, that is, laboratories that were physically isolated and nonphysically isolated from each step of library construction, respectively. NFS water: nontarget control; Target value: the proportion of SARS-CoV-2 reads in the total qualified reads. Original and new: the original and newly purchased PCR master mix; New-21 and New-24: newly synthesized primer pools containing 21 and 24 primer pairs from the original primer pool, respectively. *, $P < 0.05$ by Wilcoxon rank-sum test.

as DNA spike-ins. (Fig. 3A; Supplemental file 1: Table S3). The DNA Spike-ins harbor significant nucleotide differences from the original sequence, but the prim-binding regions of both are the same. Therefore, addition of a sufficient amount of DNA spike-ins to samples prior to library preparation reduced contaminations through competitive amplification and ensured that samples containing very low or no concentrations of virus could generate sufficient material for sequencing and analysis. Water samples supplemented with 100,000, 50,000, 10,000, and 5,000 copies of spike-ins per reaction were tested. The concentration of the library constructed with samples containing 10,000 copies of DNA spike-ins reached the lowest library concentration required for high-throughput sequencing (approximately 5 ng/μL) (Fig. 3B). Next, we tested the gradient dilutions (1, 10, and 100 copies/reaction with an undetermined cycle threshold [Ct] value, a Ct value of 33, and a Ct value of 37, respectively; Supplemental file 1: Table S4 and S5) of the SARS-CoV-2 standards supplemented with 10,000 copies of DNA spike-ins. An NTC group was used which included NFS water, the NFS-water cDNA library (hereinafter referred to as R-control), and the NFS water containing 10,000 copies of DNA spike-ins (hereinafter referred to as DNA spike-ins NTC). Three technical

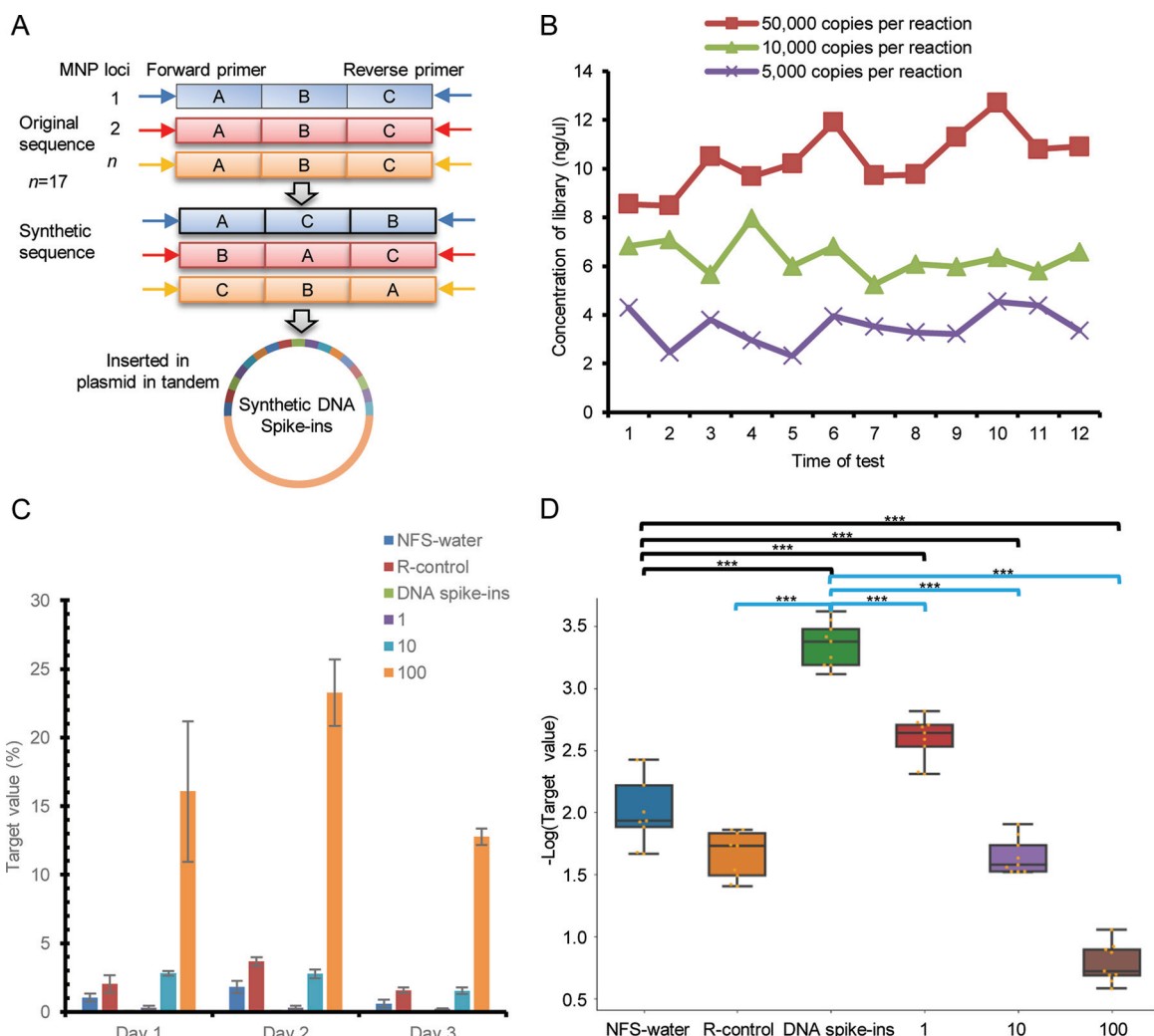

**FIG 3** Effective strategies for contamination prevention. (A) The generation of DNA spike-ins; (B) the proper number of DNA spike-in copies added to each reaction was 10,000. (C) Daily comparison of target values detected in the NTC group and SARS-CoV-2 standards. (D) Summary analysis of the target values detected in NTC group and SARS-CoV-2 standards for three consecutive days. Target value: the proportion of SARS-CoV-2 reads in the total qualified reads. Groups with $P < 0.05$ and 0.005 by Wilcoxon rank-sum test were labeled * and ***, respectively. NFS water, Nuclease-free and sterile water. R-control, cDNA library of NFS water. 1, 10, and 100, the number of SARS-CoV-2 copies/reaction.

replicates were performed daily for three consecutive days. Over the three consecutive days of testing, the DNA spike-ins NTCs always had the lowest T values of the samples tested daily (Fig. 3C). Among all the 9 replicates of each sample, T values for DNA spike-ins NTC (0.05% on mean) were significantly lower than the NFS water NTCs (1.14% on mean), R-control (2.43% on mean), and the 1, 10, and 100 copies per SARS-CoV-2 samples (0.28%, 2.37% and 17.37% on mean, respectively), indicating that with the addition of DNA spike-ins, AMP-Seq detected all positive samples and determined all negative samples as negative; namely, the diagnostic sensitivity and specificity could reach 100%. However, T values for the NFS water and R-control NTCs were significantly higher than the one copy per SARS-CoV-2 sample and lower than the 10 and 100 copies per SARS-CoV-2 sample (Wilcoxon rank-sum test, $P < 0.05$; Fig. 3D). Thus, for nucleic acid standards, the limit of detection of AMP-Seq without the addition of DNA spike-ins was 10 copies/reaction, which is approximately 10 times less sensitive than the LOD of one copy/reaction with the DNA spike-ins added.

**(ii) dUTPs and the thermolabile uracil DNA glycosylase system.** As carryover contamination is difficult to completely prevent, the most effective way to control contamination is

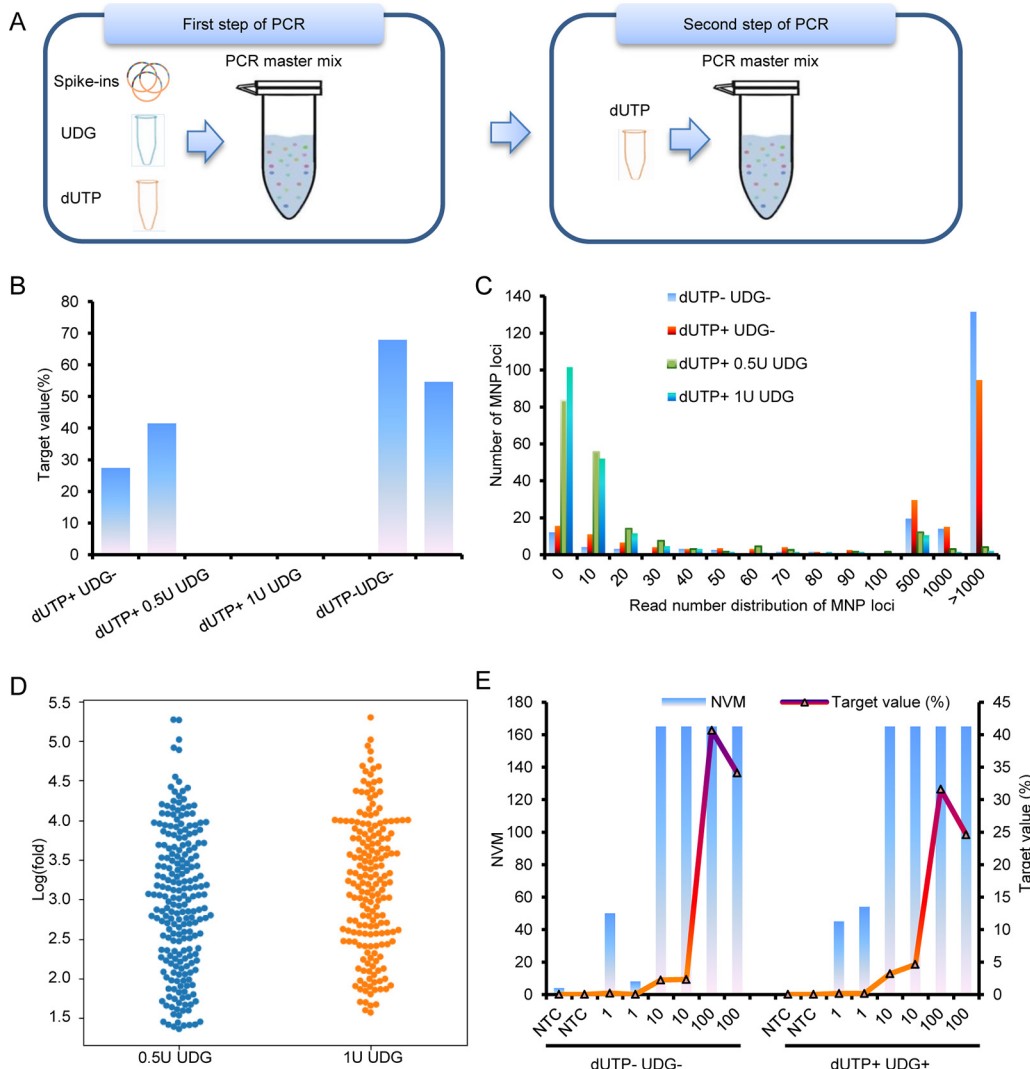

**FIG 4** Effective strategy for contamination removal. (A) The addition of dUTP/UDG system to AMP-Seq. (B, C, and D) The addition of dUTP/UDG effectively reduced the target values (B), the read number of most SARS-CoV-2 MNP loci (C), and the read number of MNP loci was reduced by at least 22-fold (D). (E) The SARS-CoV-2 standards detected with dUTP/UDG system showed a comparable number of valid SARS-CoV-2 MNP loci (NVM) and lower target values than that detected without dUTP/UDG system. Target value, the proportion of SARS-CoV-2 reads in the total qualified reads. 1, 10, and 100, the number of SARS-CoV-2 copies/reaction.

the decontamination of samples before PCR. Considering the results of previous studies, we incorporated a dUTP/UDG system into the AMP-Seq workflow by adding dUTP/UDG in the first step PCR and dUTP in the second step PCR (Fig. 4A). To mimic carryover contamination from prior amplicons, we tested the 100-copy per SARS-CoV-2 standards by supplemented dUTP with ratios of 1:1, 2:1 and 3:1 to dTTP in both steps of PCR. Samples supplemented with a dUTP to dTTP ratio of more than 2:1 produced substandard libraries; thus, a ratio of 2:1 was identified as the optimum dUTP to dTTP ratio for the multiplex PCR system. The libraries of samples with a 2:1 dUTP to dTTP ratio were then used as the templates to determine the proper concentration of UDG per reaction (Samples 88 to 95 in Supplemental file 1: Table S6). Two replicates were performed for each condition. The mean T values for the samples following the addition of 0.5 and 1U UDG were reduced to 0.16 and 0.07%, respectively, which were markedly lower than that of the samples without UDG (47.12%; Fig. 4B). Similarly, the means of numbers of valid MNP loci (NVM) detected in samples containing 0.5 and 1 U UDG were 111 and 93, respectively, which was substantially less than without UDG (180 on mean). The numbers of

reads for most valid MNP loci in samples containing UDGs were less than 10, whereas the numbers of reads in samples without UDGs were greater than 1000 (Fig. 4C). By comparing the numbers of reads for commonly detected MNP loci between samples with and without UDG, the addition of UDG reduce the number of reads of MNP loci by 22 to 199,016-fold (Fig. 4D). These results showed that the dUTP/UDG system with the ratio of dUTP/dTTP of 2:1 and 1U UDG per reaction could effectively reduce the carryover contamination.

We then tested SARS-CoV-2 standards and DNA spike-ins NTCs using AMP-Seq with and without the dUTP/UDG system to determine the influence of the dUTP/UDG system on the sensitivity and specificity of AMP-Seq. Each condition performed two replicates. In the NTCs, only 0 of $4.8 \times 10^6$ and 1 of $3.6 \times 10^6$ reads were mapped to SARS-CoV-2 with the dUTP/UDG system, while 544 of $4.4 \times 10^6$ and 0 of $4.9 \times 10^6$ reads were mapped to SARS-CoV-2 without the dUTP/UDG system, indicating that dUTP/UDG system eliminated almost all the carryover contaminations in the workflow. As a result, the T values in NTCs detected using dUTP/UDG system were lower than without dUTP/UDG system (0.00003% versus 0.01226%). The NVMs detected in SARS-CoV-2 standards with and without dUTP/UDG system were comparable. These results demonstrated that the addition of dUTP/UDG to the AMP-Seq workflow did not affect the efficiency of multiplex PCR amplification but further lowered the risk of carryover contamination, thereby strengthening the detection sensitivity and specificity of AMP-Seq (Fig. 4E, Samples 96 to 111 in Supplemental file 1: Table S6).

Thus, we established a ccAMP-Seq workflow for SARS-CoV-2 detection by adopting following measures to AMP-seq: (i) addition of the fixed copies of DNA spike-ins to samples and using DNA spike-ins as NTCs; (ii) addition of dUTP/UDG to the first PCR and dUTP to the second PCR step; and (iii) addition of a 10-minute 25℃ PCR step before the first PCR step.

**Establishment of the data analysis procedure to ccAMP-Seq.** Since DNA spike-ins was used as NTCs and added to each sample before library construction, it could be used as a benchmark to normalize a batch of samples. We reanalyzed the sequencing data of gradient dilutions of SARS-CoV-2 standards and the DNA spike-ins NTCs (Supplemental file 1: Table S2). We observed that the ratio of read numbers of SARS-CoV-2 to DNA spike-ins for the gradient dilutions of SARS-CoV-2 standards increased with the copy number of SARS-CoV-2 in the standards (Fig. 5A; Samples 40 to 87 in Supplemental file 1: Table S6). We introduced parameters, including a signal value (S value), noise value (N value), and the S to N ratio (SNR). The S value represents the ratio of read numbers of SARS-CoV-2 to DNA spike-ins of each sample, with the maximum S values of NTCs in the same batch used as the N value for each batch. We found that the S values for SARS-CoV-2 standards were at least two folds to that of NTCs. With the addition of dUTP/UDG, even greater differences in SNR observed between the NTCs and one-copy per SARS-CoV-2 samples (Samples 96 to 111 in Supplemental file 1: Table S6). NVM values have the same characteristics (Fig. 5B). Therefore, a new data analysis procedure to ccAMP-Seq was developed by taking SNR and NVM into account.

The threshold NVM and SNR values for the detection of SARS-CoV-2 by ccAMP-Seq can be determined based on the values of the one copy per SARS-CoV-2 samples (58 and $5.4 \times 10^3$ on mean, respectively). It is important to note that the SARS-CoV-2 samples containing one copy SARS-CoV-2 were SARS-CoV-2 nucleic acid standards. As the quality of nucleic acid in clinical samples is typically lower than in standards, the threshold SNR and NVM values for determining the presence of SARS-CoV-2 was set at 10 (10-fold greater than the signal value of the NTCs) and 33 (20% of all MNP loci), respectively.

Thus, the data analysis procedure to ccAMP-Seq based on the NVM and SNR values was as follows (Fig. 5C). The first step of this procedure is to determine if DNA spike-ins NTCs were present. If this was the case, we next determined if all DNA spike-ins were detectable. If all 17 spike-ins were detected in NTCs, the samples tested in the same sequencing batch were considered of adequate quality for sequencing and the presence

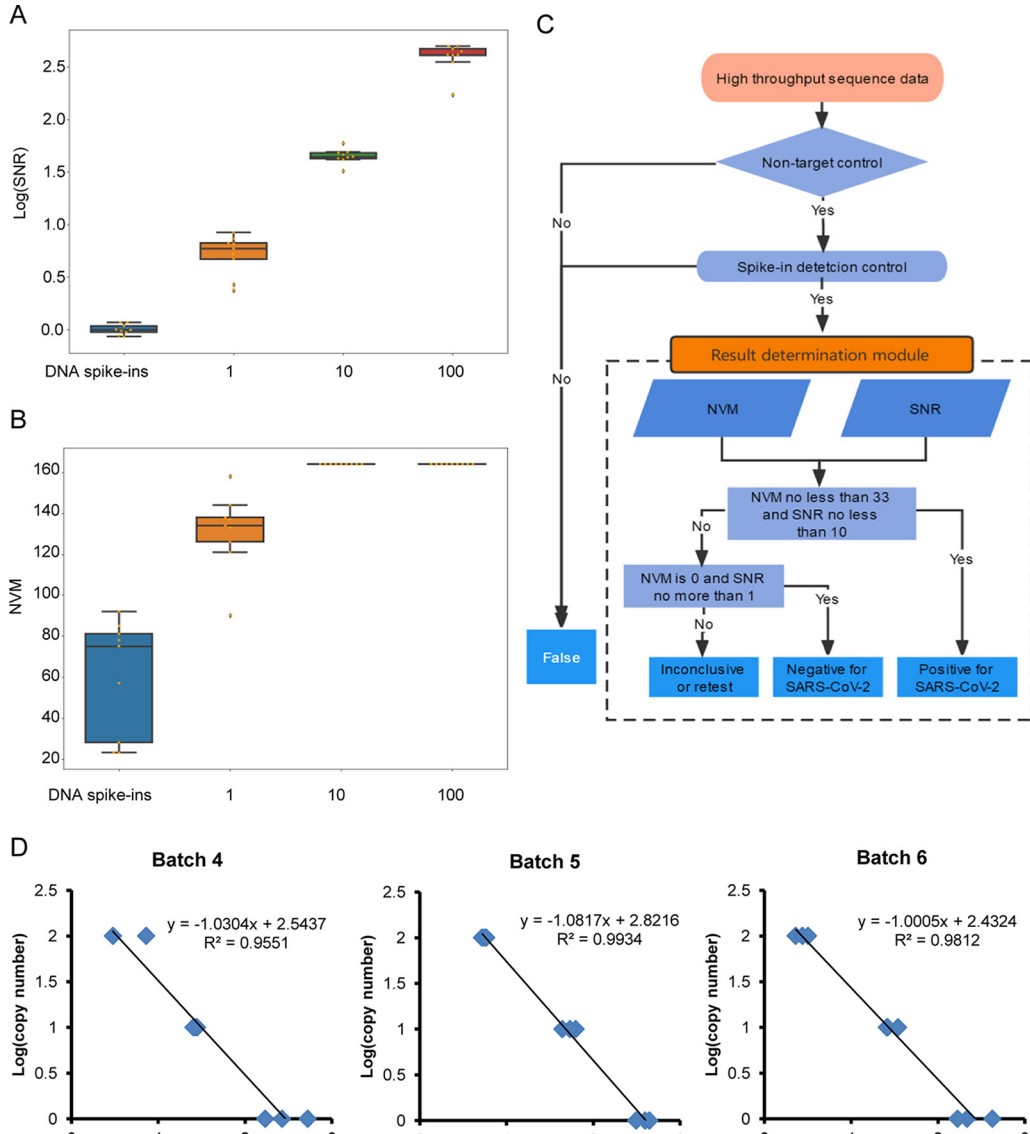

**FIG 5** Establishment of the data analysis procedure for ccAMP-Seq workflow. (A and B) The SNR (A) and NVM (B) for the gradient dilutions of SARS-CoV-2 standards were positively associated with the copies of SARS-CoV-2/reaction. (C) The flow chart of data analysis procedure of ccAMP-Seq. NVM: number of valid SARS-CoV-2 MNP loci. SNR: signal to noise ratio. 1, 10, and 100: the number of SARS-CoV-2 copies/reaction. (D) Correlation analysis between the SARS-CoV-2 copy numbers and the SARS-CoV-2 signal value (S) in samples.

of DNA spike-ins was assessed in the tested samples. When all 17 spike-ins were detected in a tested sample, the sample was entered into the result determination module to determine the presence or absence of SARS-CoV-2. If any of these steps did not meet the stipulated criteria, the test of the sample was deemed failed, and the process of the sample was ended. The rules of result determination module for determining samples in the same batch of library construction were as following: when the NVM of SARS-CoV-2 was not less than 33 and the SNR was not less than 10, the sample was considered SARS-CoV-2 nucleic acid-positive. When the NVM of SARS-CoV-2 was zero or the SNR was no more than one, the sample was considered SARS-CoV-2 nucleic acid-negative. For other cases, the sample was considered undetermined. The requirement for retesting of undetermined samples depends on the purpose of the test being performed.

**Quantification of SARS-CoV-2 by ccAMP-Seq.** To demonstrate the performance of ccAMP-Seq for SARS-CoV-2 quantification, samples of SARS-CoV-2 standards with known

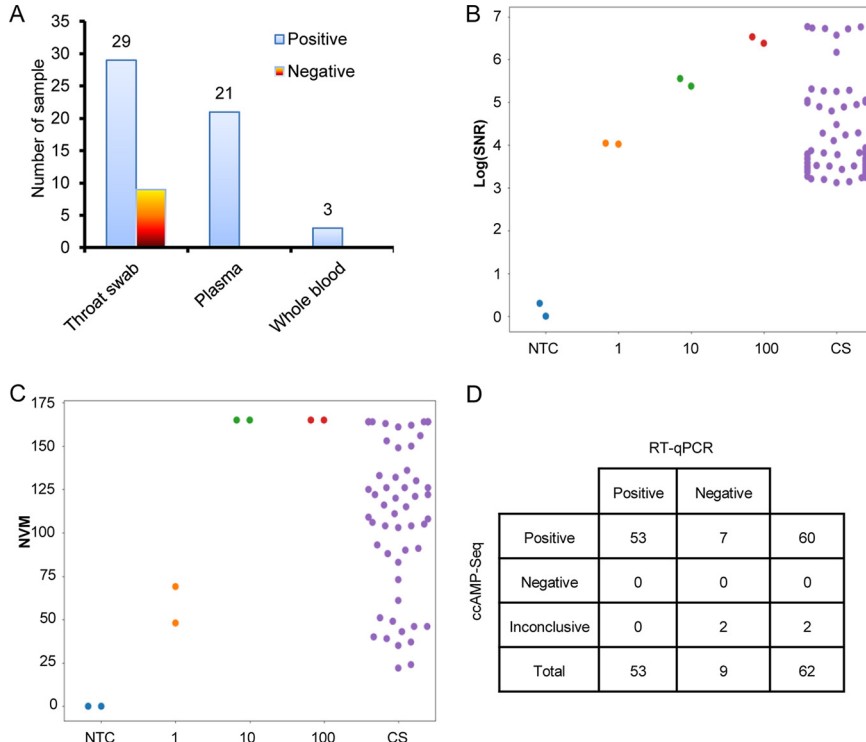

**FIG 6** The application of ccAMP-Seq on detecting SARS-CoV-2 in 62 clinical specimens. (A) The type of 62 SARS-CoV-2 clinical specimens; (B and C) SNR and NVM detected in 62 clinical samples and parallel tested gradient dilutions of SARS-CoV-2 standards. (D) A 2 × 3 contingency table comparing ccAMP-Seq to qRT-PCR for 62 clinical specimens for detecting SARS-CoV-2. SNR. signal to noise ratio. NVM, number of valid SARS-CoV-2 MNP loci. CS. clinical specimen.

copy numbers tested in the three consecutive batches were analyzed. By plotting the SARS-CoV-2 copy number relative to the S value, we observed a strong linear relationship between the log values of SARS-CoV-2 copy numbers and the log values of S values of the 10-fold dilutions of SARS-CoV-2 standards in each library construction batch ($R2 = 0.955$, 0.981, and 0.99, respectively). The results of the three consecutive batches also proved that the linear relationship was reproducible (Fig. 5D, Supplemental file 1: Table S6). Therefore, the ccAMP-Seq workflow can further quantify SARS-CoV-2 copy numbers in samples by detecting gradient dilution of samples with known viral copy numbers in parallel and generating the formula to calculate the copy number based on the S value of SARS-CoV-2.

**Application of ccAMP-Seq for clinical specimens.** ccAMP-Seq was used to test 62 clinical specimens from COVID-19 patients, including 38 throat swabs, 21 plasma, and 3 whole blood samples (Fig. 6A; Samples 112 to 173 in Supplemental file 1: Table 6). Of these, 53 samples were qRT-PCR-positive, and 9 samples were qRT-PCR-negative collected from patients before be confirmed positive. The SNR and NVM of the 62 specimens ranged from $3.8 \times 10^2$ to $4.3 \times 10^8$ and 22 to 164, respectively (Fig. 6B and 6C). According to the result determination rules of ccAMP-Seq, all of the qRT-PCR-positive samples were identified as SARS-CoV-2 nucleic acid-positive by ccAMP-Seq, whereas seven of the nine qPCR-negative samples were determined to be positive and two were undetermined (Fig. 6D). These results confirmed the higher sensitivity of ccAMP-Seq compared to qPCR. Therefore, ccAMP-Seq can be applied to a range of clinical specimens with a low risk of negative tests for SARS-CoV-2.

## DISCUSSION

In the present study, we have developed ccAMP-Seq workflow for the accurate and quantitative detection of SARS-CoV-2. The workflow combined strategies, including physical

separation of individual experimental steps, the use of pipette filter tips to reduce cross contamination, the addition of DNA spike-ins and dUTP/UDG system, and a new data analysis procedure to control contamination during amplicon sequencing workflow. DNA spike-ins has been used as internal controls and quantitative measurement throughout the sequencing workflow (18–20). In the present study, DNA spike-ins harboring the same primer-binding zones with the native sequences were used to reduce the amplification of contaminants. It should be noted that the copy number of DNA spike-in in each reaction should be optimized to a number that prevents contamination but does not consume too much sequencing data. It was approximately 10,000 copies in this study, consistent with the results of a previous study in which a microbial load of less than $10^3$ to $10^4$ cells prevented the acquisition of robust results due to abundant contamination (21). The function of dUTP/UDG system is to eliminate contamination prior to PCR. As shown in the dPCR and qPCR workflow, complete replacement of dTTP by dUTP maximizes the removal of contaminants (14, 16, 22). However, the dUTP to dTTP ratio should be carefully determined in multiplexed amplicon sequencing workflows, as we found that a ratio of more than 2:1 leaded to the creation of substandard cDNA libraries.

We have shown that the detection limit of ccAMP-Seq can be as low as one copy/reaction. It should be noted that the detection limit of one copy per reaction for ccAMP-seq is evaluated based on the detection results of SARS-CoV-2 nucleic acid standard samples containing one copy SARS-CoV-2. Because consistently reproducing copy numbers of virus in samples with extremely low viral load is a challenge to existing methods. For example, the Ct values of N and ORF1b genes in the three repeated tests for the one copy SARS-CoV-2 nucleic acid standards were ~39 or undetermined (Table S5). Therefore, the copy number of SARS-CoV-2 in the nucleic acid samples in this study was calculated based on the dilution and the original copy number of the nucleic acid standards. Compared with qPCR, ccAMP-Seq detected stable higher viral signals in these 1-copy standard samples than in the NTCs (Fig. 5A and 5B), and sufficient virus positive signals in seven qPCR-negative samples from COVID-19 patients (Fig. 6B and C), demonstrating the high detection sensitivity of ccAMP-Seq to SARS-CoV-2.

We have developed a new data analysis procedure for ccAMP-Seq. Uncertainties such as nucleic acid degradation and carryover contamination in the samples can lead to uncertain test results of samples. Therefore, the result determination module in the data analysis procedure includes three options: positive, negative, and undetermined. From our sequencing data, a small number of viral reads were still present in the NTCs tested by contamination-controlled AMP-Seq (ccAMP-Seq) (Table S6), suggesting that contamination still exists in the workflow. One of the reasons for the residue of contamination in the workflow is that in order to ensure the amplification efficiency of multiplex PCR, dUTP cannot completely replace dTTP in the system, resulting in some amplified products cannot be recognized by UDG. Another reason is that the UDG enzyme did not fully cleaved the carryover amplions. The presence of carryover contamination has the risk of causing a negative sample to be determined as an undetermined sample. Additionally, when the nucleic acid in the samples is severely degraded, the samples are possible to be determined as undetermined samples because most of the MNP loci could not be effectively amplified, resulting in insufficient positive values. In this study, two qPCR-negative samples from COVID19 patients were determined as undetermined samples by ccAMP-Seq due to the detection of less than 20% of all MNP loci. The detection rate of MNP loci in the samples was less than 20% of all MNP sites, suggesting the existence of poor nucleic acids in the samples.

The key points of our computational analysis procedure were that DNA spike-ins were used as NTC to normalize the signal values across samples, leading to the generation of S parameter for quantitative detection of pathogens in each sample. Therefore, ccAMP-Seq has advantages in the early detection and surveillance of infectious diseases due to its high sensitivity and precise quantification. Although ccAMNP-Seq detected only one pathogen in the present study, the hundreds or thousands of amplicons can target multiple pathogens in practice (3, 4, 23). For example, in our recently

formulated national standard, nine DNA viruses were detected in a single reaction (Standard no. GB/T 41895-2022). This capability of comprehensive detection of multiple pathogens makes ccAMP-Seq suitable for screening of pathogens causing similar symptoms, such as pathogens with respiratory symptoms or intestinal symptoms. The comprehensive detection facilitates precision medicine and epidemiological surveillance. In addition, ccAMP-Seq has the power to detect the genetic variation and trace the origin of infectious diseases pathogens based on the sequence of multiple markers (1, 3, 5, 24). In conclusion, ccAMP-Seq greatly improved the detection efficiency, sensitivity, specificity, accuracy, whereas it reduced the sample size and detection cost; thus ccAMP-Seq was widely applicable in pathogen-related detection scenarios.

More importantly, ccAMP-Seq is simple and economic to use. First, DNA spike-ins and dUTP/UDG systems can be preloaded in the PCR master mix for use. Second, DNA spike-ins can be produced simply, economically, and efficiently by extracting plasmid containing DNA spike-ins from transformed *E. coli*. Third, dUTP and UDG are readily available and generally cost less.

Except carryover contaminations, other categories of contaminations during high-throughput sequencing include sample-to-sample contamination (25, 26), reagent and laboratory contamination (21, 27, 28) can also occur during amplicon sequencing workflow, but ccAMP-Seq has limited effect on them. Thus, strategies such as standardize experimental operations and other kind of synthetic DNA spike-in (25) can be used to control the contamination.

**Conclusions.** Carryover contamination during amplicon sequencing workflow compromised the accuracy of the routine and high-throughput detection of dangerous pathogens, such as SARS-CoV-2, at risk. In this study, we established a contamination-control workflow and demonstrated its superiority over conventional workflow on detection accuracy, analytic sensitivity and quantification using the detection of SARS-CoV-2 as case. The developed contamination-controlled workflow was readily applicable in other amplicon sequencing workflows given its economy and simplicity.

**Ethics approval and consent to participate.** The study was approved by the Medical Ethics Committee of the Yueyang Central Hospital (No.:2022-072). Informed consent was waived based on the basis that this is a retrospective study using anonymous samples.

## MATERIALS AND METHODS

**SARS-CoV-2 RNA sample and nucleic acid standards.** RNA from clinical specimens was provided by Yueyang Central Hospital (Yueyang, China). No identifiers linking specimens to any specific individual were provided. The SARS-CoV-2 nucleic acid standards were the Certified Reference Material of 2019 Novel Corona Virus (2019-nCoV) RNA Genome purchased from the National Sharing Platform for Reference Materials (GBW [E]091099, https://www.ncrm.org.cn/Web/Ordering/MaterialDetail?autoID=20062).

**qRT-PCR assay.** The SARS-CoV-2 nucleic acid standards were diluted to 100 copies per microliter (copies/$\mu$L) in nuclease-free sterile water. The Complementary DNA (cDNA) of SARS-CoV-2 standards and clinical samples were synthesized using a CFX96 C1000 thermal cycler (Bio-Rad Laboratories) and HiScript III 1st Strand cDNA Synthesis kits (+gDNA wiper) (R312, Vazyme Biotech Co. Ltd., Nanjing, China). Dilations of the SARS-CoV-2 standards of 1, 10, and 100 copies/$\mu$L were prepared through dilution of the cDNA libraries of the SARS-CoV-2 standards according to the original concentration, and then confirmed by detecting the N and ORF1ab regions of the SARS-CoV-2 genome using qRT-PCR. The qPCR tests were performed at our laboratory using the AceQ qPCR Probe Master Mix (Q112, Vazyme Biotech Co. Ltd., Nanjing, China) on an ABI StepOnePlus real-time PCR system. qRT-PCR tests of clinical specimens were routinely performed at the hospital using 2019-nCoV nucleic test kits (Bojie Ltd., Shanghai, China).

**Design of MNP loci and synthetic DNA spike-ins.** MNP markers design in the SARS-CoV-2 genome was described in our previous study (17). Briefly, reported SARS-CoV-2 genomic sequences were compared with the genomic data of representative SARS-CoV-2 strain, Wuhan-Hu-1 (GenBank accession number: MN908947) to select regions with a high number of polymorphisms. Considering the possible low quality of nucleic acids extracted from clinical samples, genomic regions of <150 bp in length and containing ≥1SNPs were selected as candidate MNP loci. Primers for multiplex PCR amplification of the MNP loci were designed using Ion AmpliSeq Designer (https://ampliseq.com) and synthesized as primer pools. The compatibility of the multiplex primers was evaluated as the ability to detect the SARS-CoV-2 standards. For quality control of RNA of clinical specimens, 11 primer pairs for human housekeeping gene GAPDH (GenBank accession number: NM_001289746.2) amplification were designed. Finally, the 175 primers, including 164 primers for detection of 164 SARS-CoV-2 MNP loci and 11 primers for detection of human GAPDH gene were synthesized for further detection of SARS-CoV-2.

Synthetic DNA spike-ins were designed by rearranging the target sequence of 17 MNP loci with two key elements: (i) primer binding sites from the MNP loci of SARS-CoV-2; (ii) an optimized synthetic random sequence from the original sequence of MNP loci as a stuffer sequence to ensure the same length and GC content as the *in vivo* target, but had no significant homology to any known natural sequence. Finally, spike-ins consisted of 17 synthetic SARS-CoV-2 MNP loci-derived fragments. To facilitate production and preservation, the spike-ins were inserted into a plasmid.

**Library construction and high-throughput sequencing of samples.** PCR was done in two steps for library construction. Briefly, 2 $\mu$L of cDNA was amplified by the primer pool at the first step of PCR with 22 cycles, followed by a second step of 10 cycles to synthesize the sequencing libraries by introducing sequence adaptors and sample barcodes. Finally, the barcode-ligated libraries were purified and quantitated using the Qubit dsDNA HS Assay kits (Invitrogen, Thermo Fisher). The libraries with a concentration no less than 1 ng/$\mu$L (without spike-ins) and 5 ng/$\mu$L (with spike-ins) were considered qualified and mixed to a library pool. Samples of the library pool at concentrations of 40 fMol or greater were sequenced using the MGISEQ200-RS platform (MGI Tech, China) or Novaseq 6000 (Illumina, San Diego, CA) using 150 bp paired-end reads.

**Statistical analysis.** For each sample, the reads were first mapped to the SARS-CoV-2 reference genome using BWA (29). And then the number of valid MNP loci (NVM) and signal-to-noise ratio (SNR) values of each sample were analyzed. For an MNP locus, the reads covering the entire locus region were tallied. For those samples with no DNA spike-ins, MNP loci supported by at least five reads were considered valid. For the samples with addition of DNA spike-ins, the numbers of reads for each MNP were further normalized as reads per one million spike-in reads. MNP loci with at least five original reads and at least one read after normalization were considered valid MNP loci.

SNR is the signal to noise ratio. The value for the signal of SARS-CoV-2 in each sample, referred to as the *S* value, was calculated using the following formula:

$$S = \frac{n_T}{N_T}$$

in which $n_T$ and $N_T$ were representative of the read numbers mapped to SARS-CoV-2 and spike-ins, respectively, in each sample. The value for the noise, referred to as the *N* value, was the maximum S value for the nontargeting control (NTC) samples in the same batch. Statistical significance of difference was determined using Wilcoxon rank-sum test.

**Data availability.** The data sets supporting this study's findings are openly available in the NCBI BioProject database (https://www.ncbi.nlm.nih.gov/bioproject/) at accession number PRJNA916278.

## SUPPLEMENTAL MATERIAL

Supplemental material is available online only.

**SUPPLEMENTAL FILE 1**, XLSX file, 0.1 MB.

## ACKNOWLEDGMENTS

We have no conflicts of interest to declare.

This work was supported by the Wuhan Science and Technology Project (Grant No. 2020020601012234), Standardization Administration of China (Grant No. 20201830-T-469), Key R&D Projects of Science and Technology Department of Hubei Province (Grant No. 2020BCA090), Natural Science Foundation of Hubei Province (Grant No. 2021CFB563).

Lifen Gao: conceptualization, data curation, formal analysis, investigation, visualization, writing-original draft, writing-review and editing, funding acquisition. Lun Li and Bin Fang: data curation, formal analysis, methodology, software. Zhiwei Fang: formal analysis, methodology. Yanghai Xiang and Min Zhang: conceptualization, resources. Junfei Zhou, Huiyin Song, Tiantian Li: investigation, validation, visualization. Lihong Chen: funding acquisition, visualization. Huafeng Xiao, Renjing Wan: investigation, validation. Yongzhong Jiang: conceptualization, funding acquisition, project administration, writing-review and editing. Hai Peng: conceptualization, funding acquisition, project administration, supervision, writing-review and editing.

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
