## [Reviewer comments · Microbiology Spectrum]

Microbiology Spectrum

Carryover contamination-controlled amplicon sequencing workflow for accurate qualitative and quantitative detection of pathogens: a case study on SARS-CoV-2

Lifen Gao, Lun Li, Bin Fang, Zhiwei Fang, Yanghai Xiang, Min Zhang, Junfei Zhou, Huiyin Song, Tiantian Li, Lihong Chen, Huafeng Xiao, Renjing Wang, Yongzhong Jiang, and Hai Peng

Corresponding Author(s): Hai Peng, Jiangnan University

Review Timeline:

Submission Date:	January 13, 2023
Editorial Decision:	February 11, 2023
Revision Received:	March 20, 2023
Accepted:	April 2, 2023

Editor: Frederick S. Kibenge

Reviewer(s): Disclosure of reviewer identity is with reference to reviewer comments included in decision letter(s). The following individuals involved in review of your submission have agreed to reveal their identity: Hongping Wei (Reviewer #1)

Transaction Report:

DOI: <https://doi.org/10.1128/spectrum.00206-23>

February 11, 2023

Prof. Hai Peng
Jiangnan University
Institute for system biology
Wuhan
China

Re: Spectrum00206-23 (Carryover contamination-controlled amplicon sequencing workflow for accurate qualitative and quantitative detection of pathogens: a case study on SARS-CoV-2)

Dear Prof. Hai Peng:

The manuscript has been reviewed by one reviewer. I have also reviewed this manuscript. The reviewer's comments must be addressed in the revised manuscript.

Link Not Available

Sincerely,

Frederick S. Kibenge

Journals Department
Reviewer comments:

Reviewer #1 (Comments for the Author):

The manuscript described a ccAMP-Seq workflow to reduce the carryover contaminations during AMP-seq, which is important to improve accuracy of the method. But I have a few concerns on the results:

1. While measures such as using filter tips and physical isolation to avoid cross contamination, dUTP/uracil DNA glycosylase system to digest the carryover contaminations are understandable, it is tricky to know that synthetic DNA spike-ins could reduce the carryover contamination. As the authors mentioned, DNA spike-ins could compete with contaminations and quantify SARS-CoV-2. Could the competition mask the amplification of the low viral genes really present in the sample? The authors should

provide more explanation on the effects of synthetic spike-ins.

2. On analysis of the sequencing data, the authors mentioned that "When the NVM of SARS-CoV-2 was no more than zero or the SNR was no more than one, the sample was considered SARS-CoV-2 nucleic acid-negative. For other cases, the sample was considered undetermined. The requirement for retesting of undetermined samples depends on the purpose of the test being performed." (lines 333-336). What does "no more than zero" mean? Could NVM be negative? I think here using "Zero" is better. Then as the authors indicated there will be a gray area "undetermined" between Negative and Positive. Under which purposes the "undetermined samples" should be retested? And which kind of samples or conditions could generate the results of "undetermined"? Carryover contaminations or low viral mass in the samples?

3. The detection limit of ccAMP-seq is 1 copy/reaction, which is really amazing if it is true. Could it mean that the ccAMP-seq will give the positive result even if there are only one copy in the reaction? If so, I do not think there will be any sample with "undetermined" result since there is only zero below one copy and zero copy should generate negative result. Hope the authors could recheck their data or explain the limit of detection further.

Staff Comments:

Preparing Revision Guidelines

Please return the manuscript within 60 days; if you cannot complete the modification within this time period, please contact me. If you do not wish to modify the manuscript and prefer to submit it to another journal, please notify me of your decision immediately so that the manuscript may be formally withdrawn from consideration by Microbiology Spectrum.

Dear editor(s),

We would like to express our sincere appreciations of your and the reviewers' professional comments concerning our article. These comments are all valuable and helpful for improving our article. This is the point-by-point responses to the issues raised by the reviewers. Changes from our original submission were all highlighted using the modified mode.

Reviewer comments:

Reviewer #1 (Comments for the Author):

The manuscript described a ccAMP-Seq workflow to reduce the carryover contaminations during AMP-seq, which is important to improve accuracy of the method. But I have a few concerns on the results:

1. While measures such as using filter tips and physical isolation to avoid cross contamination, dUTP/uracil DNA glycosylase system to digest the carryover contaminations are understandable, it is tricky to know that synthetic DNA spike-ins could reduce the carryover contamination. As the authors mentioned, DNA spike-ins could compete with contaminations and quantify SARS-CoV-2. Could the competition mask the amplification of the low viral genes really present in the sample? The authors should provide more explanation on the effects of synthetic spike-ins.

Response:

Dear reviewer, thank you for the valuable questions. Our response to each question is as follows.

For question: Could the competition mask the amplification of the low viral genes really present in the sample?

One of the main purposes of adding DNA spike-ins was to help samples containing very low or no concentrations of virus generate sufficient material for sequencing and analysis. We have tested non-target controls (NTCs) consisting of nuclease-free and sterile water (NFS water) and SARS-CoV-2 standards containing one copy of SARS-CoV-2 using AMP-Seq with and without the addition of DNA spike-ins. The libraries constructed without DNA spike-ins had a substandard concentration, for example, < 5 ng/μl in our study, which was insufficient for high throughput sequencing. On the contrary, those with DNA spike-ins were eligible for high throughput sequencing, and all of SARS-CoV-2 standards containing one copy of SARS-CoV-2 were tested SARS-CoV-2 positive based on the sequencing data (Supplementary Table S6).

One of the reasons that one copy of SARS-CoV-2 can be detected instead of being masked by the DNA spike-ins is that the SARS-CoV-2 nucleic acid standards usually

had an intact genome thus all or most of the 164 primer pairs can simultaneously amplify the genomes of SARS-CoV-2, while only 17 of them can also target DNA spike-ins. In our study, SARS-CoV-2 nucleic acid standards containing one copy SARS-CoV-2 were tested nine times. The number of MNP loci detected at each test ranged from 93 to 158. The result demonstrated that the amplification of DNA spike-ins does not conceal the detection of low copy SARS-CoV-2 samples.

2. On analysis of the sequencing data, the authors mentioned that "When the NVM of SARS-CoV-2 was no more than zero or the SNR was no more than one, the sample was considered SARS-CoV-2 nucleic acid-negative. For other cases, the sample was considered undetermined. The requirement for retesting of undetermined samples depends on the purpose of the test being performed." (lines 333-336). What does "no more than zero" mean? Could NVM be negative? I think here using "Zero" is better. Then as the authors indicated there will a gray area "undetermined" between Negative and Positive. Under which purposes the "undermined samples" should be retested? And which kind of samples or conditions could generate the results of "undermined" ? Carryover contaminations or low viral mass in the samples?

Response:

Dear reviewer, thank you very much for the valuable questions. We responded individually and included the responses in the Result and Discussion sections of the manuscript (L334, Result section; L400-L419, Discussion section).

For question: What does "no more than zero" mean? Could NVM be negative? I think here using "Zero" is better.

We have replaced it with "Zero" as suggested (L334, Result section).

For question: which kind of samples or conditions could generate the results of "undermined"? Carryover contaminations or low viral mass in the samples?

Carryover contaminations and severe nucleic acids degradation may lead to undetermined detection results of samples.

From our sequencing data, a small number of viral reads were still present in the NTCs tested by contamination-controlled AMP-Seq (ccAMP-Seq) (Supplementary Table S6), suggesting that contamination still exists in the workflow. The dUTP/thermolabile uracil DNA glycosylase (UDG) system was used to eliminate contaminations in the ccAMP-Seq workflow by supplementing PCR reaction mixtures with dUTP and UDG. The carryover amplicons were cleaved by UDG enzyme before PCR reactions at the dUTP nucleotides. One of the reasons for the residue of contamination in the workflow is that in order to ensure the amplification efficiency of multiplex PCR, dUTP cannot completely replace dTTP in the system, resulting in some amplified products may not contain uracil that can be recognized by UDG.

Another reason is that the UDG enzyme did not fully cleaved the carryover amplicons. The presence of carryover contamination has the risk of causing a negative sample to be determined as an undetermined sample.

Referring to our response to the reviewer's first comment, samples with low viral concentrations will still be tested positive as long as the nucleic acid quality is qualified. But when the nucleic acid in the samples is severely degraded, the samples are possible to be determined as undetermined samples because most of the MNP loci could not be effectively amplified, resulting in insufficient positive values. In our study, two qPCR-negative samples from COVID19 patients were determined as undetermined samples by ccAMP-Seq due to the detection of less than 20% of all MNP loci (L400-L419, Discussion section).

For question: Under which purposes the "undetermined samples" should be retested? Carryover contamination of negative samples and nucleic acid degradation of positive samples may lead to undetermined detection results of samples. For dangerous infectious pathogens, such as SARS-CoV-2, missing positive samples raises many questions. In such cases, we recommend retesting any undetermined samples.

In our experience, retest of undetermined samples caused by carryover contamination under strict contamination control conditions will often yield different detection results because the probability of the same contamination level happened between the two repeated tests is very low. The real trouble is that the undetermined samples caused by nucleic acid degradation, and the retest result is often still undetermined. In such cases, resampling is recommended.

3. The detection limit of ccAMP-seq is 1 copy/reaction, which is really amazing if it is true. Could it mean that the ccAMP-seq will give the positive result even if there are only one copy in the reaction? If so, I do not think there will be any sample with "undetermined" result since there is only zero below one copy and zero copy should generate negative result. Hope the authors could recheck their data or explain the limit of detection further.

Response:

Dear reviewer, thank you for your valuable questions and advices.

For the detection limit of ccAMP-seq, our description in the manuscript was not rigorous enough. We have revised the description about the detection limit of ccAMP-seq in the Results and Discussion section of the manuscript as follows (L248 and L315 in Results section; L387-L399 in Discussion section).

The detection limit of one copy per reaction for ccANM-Seq is evaluated based on the detection results of SARS-CoV-2 nucleic acid standard samples containing one copy SARS-CoV-2. Because consistently reproducing copy numbers of virus in samples

with extremely low viral load is a challenge to existing methods. For example, the cycle threshold (Ct) values of N and ORF1b genes in the three repeated tests for the one copy SARS-CoV-2 nucleic acid standards were ~39 or undetermined (Supplementary Table S5). Therefore, the copy number of SARS-CoV-2 in the nucleic acid samples in this study was calculated based on the dilution and the original copy number of the nucleic acid standards. Compared with qPCR, ccAMP-Seq detected stable higher viral signals in these 1-copy standard samples than in the NTCs (Fig. 5A and 5B), and sufficient virus positive signals in seven qPCR-negative samples from COVID-19 patients (Fig. 6B and 6C), demonstrating the high detection sensitivity of ccAMP-Seq to SARS-CoV-2.

Similarly, we used ccAMP-Seq to detect gradient dilutions of nucleic acid standards for herpes simplex virus type 1 (HSV-1), herpes simplex virus type 2 (HSV-2), and Epstein-Barr virus (EBV), respectively (data unpublished). The detection limit of ccAMP-Seq for HSV-1 and EBV also reached one copy/reaction. These results confirmed the ultra-high detection sensitivity of ccAMP-Seq for samples with high quality nucleic acids.

However, due to the multiple steps of sampling, storage, transportation and nucleic acid extraction, the quality of nucleic acid in clinical samples is usually not comparable to that of the standards, so the detection limit of ccAMP-Seq cannot be generalized. We have revised the description of the detection limit in the Results and Discussion section (L248 and L315 in Results section; L387-L399 in Discussion section).

Referring to our response to the second comment of the reviewer, uncertainties such as nucleic acid degradation and carryover contamination can lead to uncertain test results of samples. Therefore, we kept the undetermined option in the result determination module of ccAMP-Seq.

April 2, 2023

Prof. Hai Peng
Jiangnan University
Institute for system biology
Wuhan
China

Re: Spectrum00206-23R1 (Carryover contamination-controlled amplicon sequencing workflow for accurate qualitative and quantitative detection of pathogens: a case study on SARS-CoV-2)

Dear Prof. Hai Peng:

Your manuscript has been accepted, and I am forwarding it to the ASM Journals Department for publication. You will be notified when your proofs are ready to be viewed.

Sincerely,

Frederick S. Kibenge
Editor, Microbiology Spectrum
